A zero-trust based scheme for detecting illegal terminals in the Internet of Things of smart grid

Zhu Hongyu 1 2
Tian Jianwei 1 2
Chen Qian 3
Tian Zheng 1 2
Luo Weiqiang 3
Li Mingguang 4 Limg@hnu.edu.cn
1 State Grid Hunan Electric Power Company Limited Information and Communication Company , Changsha, Hunan , China
2 Hunan Key Laboratory for Internet of Things in Electricity , Changsha, Hunan , China
3 State Grid Hunan Electric Power Co., Ltd. , Changsha, Hunan , China
4 College of Computer Science and Electronic Engineering, Hunan University , Changsha, Hunan , China
Coelho Paulo Jorge
Electronic publication date: 2025 Mar 7
Publication date: 2025
Volume: 11
Electronic Location ID: e2736
Received 2024 Aug 2; Accepted 2025 Feb 6
Copyright: © 2025 Zhu et al.
Copyright year: 2025
Copyright holder: Zhu et al.
License: This is an open access article distributed under the terms of the Creative Commons Attribution License, which permits unrestricted use, distribution, reproduction and adaptation in any medium and for any purpose provided that it is properly attributed. For attribution, the original author(s), title, publication source (PeerJ Computer Science) and either DOI or URL of the article must be cited.
License URL: https://creativecommons.org/licenses/by/4.0/

Keywords: IoT terminal, Zero-trust, Smart grid, Illegal equipment detection

Funding: State Grid Hunan Electric Power Co., Ltd Research Project 5216A8220005 Hunan Key Laboratory for Internet of Things in Electricity, P. R. China 2019TP1016 This work is supported by State Grid Hunan Electric Power Co., Ltd research project No. 5216A8220005 and Hunan Key Laboratory for Internet of Things in Electricity, P. R. China No. 2019TP1016. The funders had no role in study design, data collection and analysis, decision to publish, or preparation of the manuscript.

==============================
In recent years, the Internet of Things (IoT) for electricity has faced a series of new challenges. Attackers use a compromised terminal as a springboard to enter the network, steal data, issue malicious commands, and cause great harm. In order to combat the threat of compromised terminals, this article proposes a zero-trust based detection scheme for illegal terminals, based on the principle of “never trust, always verify” security mechanism. Firstly, the detection scheme uses the state secret SM9 secret system to authenticate the access device. Then, it proposes a continuous trust evaluation based on the centroid drift trust algorithm on the characteristics of the traffic of the input device. Finally, it generates a real-time access policy by the access control engine to achieve a dynamic access policy. Finally, the access control engine generates real-time access policies to achieve dynamic access control. Experimental results show that the designed system has a high security detection accuracy and can effectively deal with the threat of compromised terminals.

Introduction

As an important infrastructure concerning international livelihoods, power systems have always been a key target of cyber attacks (Feng et al., 2022). In order to prevent network security threats and detect and intercept illegal devices in a timely manner, the main measures currently adopted include security partitioning, horizontal isolation and vertical encryption. In the new power system, the communication network is built based on the “cloud, pipe, edge and end” architecture. Figure 1 shows the communication process. Firstly, the intelligent terminal sends out an access request. Then, the edge Internet of Things (IoT) agent obtains the access request and carries out identity authentication for the device. When the device identity authentication is legal, the IoT agent establishes secure communication with the IoT management platform through the wireless private network and performs relevant services until the intelligent device sends out the communication end signal. The core of the current communication mechanism is “one-time authentication, continuous trust”.

Figure 1 The communication process.

However, with the increase of new power system access devices, the communication frequency between the information management platform and intelligent terminals is getting higher and higher, which leads to network attacks such as illegal access, data tampering and operation overstepping (Zou et al., 2021), and poses a serious challenge to the security and reliability of the power system data network.

Currently, many scholars at home and abroad are committed to the research of illegal device detection in power IoT. However, most of them are based on “once authenticated, always trusted”, which has serious security risks. To solve this problem, we introduce zero trust. The generic model of zero-trust architecture (Syed et al., 2022; Fernandez & Brazhuk, 2024) mainly includes three parts: authentication, dynamic trust evaluation and access control. It follows the concept of ‘never trust, continuous authentication’ to continuously authenticate and evaluate the security of access devices, to improve system security. Specifically, this article proposes an illegal device and other detection scheme based on zero-trust architecture. The proposed scheme aims to authenticate all the accessed smart devices and conduct continuous trust assessment to achieve continuous supervision of sessions and dynamic access control.

Related work

Unified identity management platform

Currently, many scholars at home and abroad are committed to the research of illegal device detection in power IoT: Fernandez & Brazhuk (2024) propose a power IoT smart terminal security protection technology to detect illegal power assets by analysing base station location position and device fingerprints; Hussain, Farooq & Ustun (2020) propose an association data authentication encryption algorithm to achieve the integrity and confidentiality of Generic Object Oriented Substation Event (GOOSE) messages; Chen et al. (2020) propose an integrated terminal identity authentication technology, which combines the collected user identity information with the device fingerprint to form a unique identifier that cannot be replicated, and achieves accurate detection of the legitimacy of power terminals; La, hao & Zhang (2022) propose a power terminal security authentication scheme based on SM9 threshold signatures, which achieves lightweight verification of the legitimacy of power terminals; Yang et al. (2023) propose an edge computing-oriented power terminal lightweight authentication protocol for edge computing, which reduces computation and communication overheads on the basis of effective verification of terminal legality.

The above research on detecting illegal devices in power IoT ensures the legitimacy of the access Device. However, the nature of this security authentication mechanism is “once authenticated, always trusted”, which has serious security risks.

Current research status of electric power network security solutions based on zero trust architecture

The zero-trust concept was originally born in the Jericho Forum (2020) and was formally introduced by John Kindervag in 2010. The generic model of zero-trust architecture mainly consists of authentication and management of access subjects, dynamic trust analysis and evaluation, and secure access decision and control.

In recent years, the application of zero-trust architecture has been studied in the field of power system security, but has not yet been applied on the ground. Table 1 shows the comparison of existing zero trust solutions. Liu et al. (2021) propose proposes a zero-trust-based power grid security protection model, which mainly contains trusted access agent, dynamic trust level assessment and security access control. It proposes a trust assessment algorithm that calculates the trust level based on the deviation between the feature information and the benchmark value in the trust level assessment module, but it does not explain the method of obtaining the benchmark value. Liu et al. (2020) propose a zero-trust-based energy Internet security protection architecture, which mainly includes four aspects: terminal access control, dynamic authorisation management, unified identity management and security as a service. It proposes a method of calculating the current trust measure value based on the historical trust level and risk level in the terminal access control part, but does not state the calculation method of the risk level of the key factors. Jiang et al. (2023) propose proposes an electric power IoT security access method based on the zero-trust architecture, which mainly includes the IoT terminal identity management and authentication and dynamic access control. It proposes a continuous trust assessment method based on the ratio of normal and abnormal events in the dynamic access control module, but it does not state the assessment criteria of whether the events are abnormal or not.

Table 1 The comparison of existing zero trust solutions.

Authors	Idea	Deficiencies	
Liu et al. (2021)	A trust assessment algorithm that calculates the trust level based on the deviation between the feature information and the benchmark value in the trust level assessment module.	Do not explain the method of obtaining the benchmark value	
Liu et al. (2020)	A method of calculating the current trust measure value based on the historical trust level and risk level in the terminal access control part.	Do not state the calculation method of the risk level of the key factors.	
Jiang et al. (2023)	A continuous trust assessment method based on the ratio of normal and abnormal events in the dynamic access control module.	Do not state the assessment criteria of whether the events are abnormal or not.	
Ours	Focuses on three aspects: smart device authentication, continuous trust assessment method for sessions, and dynamic access control.	Cope with the insider threat posed by the out-of-touch terminal.	

For this reason, in the illegal terminal detection scheme based on zero-trust architecture proposed in this article, it focuses on three aspects, namely, smart device authentication, continuous trust assessment method for sessions, and dynamic access control, in order to cope with the insider threat posed by the out-of-touch terminal.

Detecting illegal devices in power iot based on zero trust architecture

In the new power system, the communication network is built based on the “cloud, pipe, edge and end” architecture. However, the internal environment of the power system is not completely trustworthy, and the attacker can bypass the border network security defence system by virtue of the “legitimate identity” after invading the power terminal, invade the internal power network, and carry out the attack behaviours such as forging false data and sending malicious control commands.

Therefore, this article proposes a detection scheme for illegal terminals based on zero-trust architecture. Zero trust Zero trust is a network security policy used to authenticate every user, device, application and transaction. It follows the principle of ‘never trust, always authenticate’, breaks the original static and default trust, and has become an effective means to combat insider threats (Jiang et al., 2019). Zero trust is an integration of security concepts and security policies. On the one hand, the security concept requires all accesses to be verified and authorised, dynamically verified, and never trusted. On the other hand, the security policy is the access control relationship in information security to access the subject and object, which changes dynamically with the environment of the subject and object. The proposed approach is based on the idea of zero trust, which does not follow any implicit trust mechanism and always suspects the accessing device. In addition, it generates a secure access policy by authenticating and continuously trusting the smart device requesting access, and enforces the access policy in real time for dynamic control, so as to detect illegal terminals in time and ensure communication security.

The structure of the IoT scheme for detecting illegal devices based on zero-trust architecture is shown in Fig. 2. The proposed method has two planes, data and control. The control plane contains three core functional platforms: unified identity management platform, continuous trust assessment platform and access control engine. The unified identity management platform undertakes the function of authenticating the data reliability and identity legitimacy of the smart device requesting access; the function of the continuous trust assessment platform is to continuously calculate the trust level of the access device during the session and assess the security risk; the function of the access control engine is to formulate the access policy and execute it in real time to achieve dynamic access control. The data plane mainly contains access subject, device agent, and access object. The access subject is the issuer of the access request; the device agent is a new module of the edge IoT agent, which receives the access request from the device in real time on the smart device side; the access object is the data and other resources that can be accessed.

Figure 2 Structure of illegal terminal detection scheme for IoT based on zero-trust architecture.

The working mechanism of the proposed method is shown as Fig. 3. first, in the data plane, an intelligent device such as an inspection robot issues an access request as an access object, and the device agent obtains this request and forwards it to the control plane. Then, in the control plane, the access control engine receives the access request and establishes communication with the unified identity management platform and the continuous trust assessment platform, respectively, to obtain the identity authentication results and the trust assessment results and formulate the real-time access policy. Finally, according to the access policy, the local device agent establishes trusted communication with the access guest. Based on the zero-trust concept, the system continuously evaluates the trust-worthiness of the accessing device during the session and generates real-time access policies and dynamically executes them based on the evaluation results.

Figure 3 The working mechanism of our method.

Unified identity management platform

Identity authentication technology is an effective solution to guarantee information security. Currently, most of the device authentication in power IoT is based on the traditional public key infrastructure method. These approaches create a certificate for each end device and exchanges certificates during authentication to check the legitimacy of the end identity. However, they have cumbersome certificate exchanges, making the authentication and management system very complex. With the access of a large number of terminal devices, it is difficult for them to respond quickly to the authentication request of each terminal.

For this reason, the proposed method establishes a unified identity management platform, verifies the data reliability and identity legitimacy of smart terminals based on identity identification, and manages user identity information in a unified manner. The identity management platform consists of a user identification information database, a user public key database and a digital signature verification algorithm module. Among them, the identity information of smart devices is collected and stored in advance to the user identity information database; the user public key is generated and issued by the third-party trusted platform according to the user identity information. When the smart device sends out an access request, the identity management platform receives the uplink access data forwarded by the device agent and access control engine, verifies the reliability of the digital signature and the legitimacy of the identity, and uniformly stores and man-ages the device identity information.

The identity authentication mechanism is shown in Fig. 4, and the specific steps are as follows:

Figure 4 Unified identity authentication and management.

1. Third-party trusted platform key generation and management. The third-party trusted key management platform, i.e., the key generation centre, undertakes key generation and management functions, generates the initial master private key and master public key through the random number generator, and generates the user’s private key and user’s public key and sends them out according to the user’s request for key generation, combining the user’s identity with the master private key;

2. The local device agent obtains the access request from the smart device and performs digital signature. After the intelligent device sends an access request, the local device agent obtains the request and requests the user private key from the key generation centre according to the corresponding user identification (i.e., device number), then digitally signs the data requested for upstream transmission according to the user private key, and finally upstream transmits the digital signature and service data to the access control engine;

3. The access control engine forwards the digital signature and service data to the unified identity management platform;

4. The unified identity management platform verifies the identity of the smart device. The unified identity management platform requests the key generation centre to generate a public key according to the corresponding user identification and stores it in the user public key database, and then verifies the reliability of the digital signature in the digital signature verification algorithm module, i.e., verifies the authenticity and integrity of the received data and checks whether the identity of the device that sends this data is legal.

The authentication mechanism is designed as an embedded module, which facilitates flexible replacement of the authentication scheme. We select SM9 (La, hao & Zhang, 2022; Yuan & Cheng, 2016) cryptosystem as an authentication module embedded in the authentication platform. The core of SM9 is an asymmetric algorithm based on binary linear pairs, which has higher computational complexity and improves the level of security protection of identity information; at the same time, the system calculates and generates the user key pairs according to the user identification, the master public key, the master private key and the public parameter, and does not need to apply for and exchange digital certificates, which reduces the complexity of the key management and use in the identity authentication. The embedded module approach makes the selection of authentication schemes more flexible. In practical applications, the authentication scheme can be flexibly changed according to user requirements, such as RSA (Burkhardt et al., 2023), AES (Aharoni et al., 2023), etc., to meet specific encryption standards.

Continuous confidence assessment platform

The trust assessment mechanism is an effective programme for controlling access in network security. Currently, most smart substations follow the secure communication mechanism of “one-time authentication, continuous trust”. This mechanism establishes communication between security-authenticated smart devices and the IoT management platform, and always trusts the access device until the end of communication. However, the current internal environment of the power system is not completely reliable, smart devices that have already established communication may be exploited. In response to such malicious behaviours such as carrying out data tampering and overstepping access rights, it is difficult for traditional trust mechanisms to detect abnormalities in a timely manner.

In this regard, the proposed method is based on the concept of zero trust, following the “never trust, always verify” trust mechanism (BILGERB, 2013), and establishes a continuous trust assessment platform to continuously assess the trustworthiness of access devices. The goal of trust assessment is to detect whether there is any abnormality in the current device, and the judgement criterion is whether the device deviates significantly from the normal communication behaviour during the continuous assessment process. To assess the communication behaviour of the device, the communication characteristic value of the device is extracted. In normal service communication, the device transmits data upstream at a stable frequency, the ratio of nodes receiving packets and sending packets is constant, the network delay does not change much, and the difference in device communication time is small. Therefore, the node packet delivery rate, average delay, and average communication time are extracted as communication eigenvalues. The communication eigenvalues are defined as shown in Eqs. (1)–(3), the packet delivery rate Pratio denotes the ratio of received packets to sent packets per unit of time T, the average delay Pdelay denotes the average value of service communication delay per unit of time T, and the average communication time Pmessage denotes the average value of communication time per unit of time T.

(1) Pratio=∑i=1TReci∑i=1TSendi

(2) Pdelay=1T∑i=1T⁡Delayi

(3) Pmessage=1T∑i=1T⁡Mesi.

The trust evaluation mechanism calculates the deviation of the feature vector of the current device from the other devices in the current unit of time based on the communication feature vector P( Pratio, Pdelay, Pmessage), and then calculates the deviation of the communication feature vector between the current device and its history time. If there is a significant deviation in the communication feature vector of the current device, it indicates that its communication behaviour is abnormal and the device is untrustworthy.

In this article, based on the idea of mean drift (Okade & Biswas, 2013), we propose a trust assessment method based on centroid drift to build a dynamic detection model for the data, which does not need to make trade-off judgements on the data when building the model, and overcomes the influence of outliers. The core idea of the algorithm is to replace the communication feature vector P of a device with the centroid P′ ( Pratio′, Pdelay′, Pmessage′) of its k-nearest-neighbours’ feature vectors, and then compute the feature vector move distance, and finally compute the device trust value based on the move distance. Where the centroid of the feature vector of the nearest neighbour of device k is calculated as shown in Eqs. (4)–(6).

(4) Pratio′=max(Pratioi)+min(Pratioi)2,i=1,2,…,k

(5) Pdelay′=max(Pdelayi)+min(Pdelayi)2,i=1,2,…,k

(6) Pmessage′=max(Pmessagei)+min(Pmessagei)2,i=1,2,…,k.

The distance travelled when the device communication vector P is replaced with the centre point P′ is calculated as shown in Eq. (7).

(7) D=(ΔPratio)2+(ΔPdelay)2+(ΔPmessage)2

ΔPratio=Pratio′−Pratio

ΔPdelay=Pdelay′−Pdelay

ΔPmessage=Pmessage′−Pmessage.

The confidence level is calculated as shown in Eqs. (8) and (9), where εδ is used to describe the empirical parameters.

(8) T=11+e−δ+DΔPratio=Pratio′−Pratio

(9) δ=1n∑i=1n⁡Di+εδ,i=1,2,…,n.

At this point, the value of trust degree is in the range of (0,1), and the higher the outlier value of the device, the higher the trust degree. When the trust degree is higher than 0.5, then this device can be trusted, otherwise the device cannot be trusted.

The algorithm for continuous trust assessment based on centroid drift is shown below:

1. Calculate the current trustworthiness value of the smart device based on the communication feature values. The platform obtains the incoming smart device access request from the access control engine, and extracts the communication feature values of the current access device according to Eqs. (1)–(3), which are noted as Pratio∗, Pdelay∗ and Pmessage∗, respectively;

Obtain the communication feature values of the current n access devices, noted as Pratioi,Pdelayi,Pmessagei(i=1,2,…,n);

Find the k-nearest neighbours of each device and compute the centroid pi′(Pratio′,Pdelay′,Pmessage′) of the k-nearest neighbours. In each iteration, the k-nearest neighbours of the ith device are found and the centroid is calculated according to Eqs. (4)–(6). Where k is selected by choosing a value proportional to the amount of data n (Yang, Rahardja & Fränti, 2021) as shown in Eq. (10):

(10) k=2×[5×log⁡(n)2]+1.

The device communication point pi′(Pratioi′,Pdelayi′,Pmessagei′)(i=1,2,…,n) is replaced with the device’s k-nearest-neighbour centroid pi′(Pratioi,Pdelayi,Pmessagei). Calculate the current device P∗(Pratio∗, Pdelay∗,Pmessage∗) k nearest neighbour centroid p∗′(Pratio∗′, Pdelay∗′,Pmessage∗′) according to Formulas (4)–(6). Finally the outliers D∗ of the communication point P∗ are calculated according to Eq. (7);

Calculate the real-time trust value T1 of the current device according to Eqs. (8) and (9).The value range of trust is [0,1], and the larger the outlier value of the device is, the higher the trust is. When the outlier value is equal to the threshold value, the trust degree value is 0.5, i.e., 0.5 is the trust degree threshold value, and the device can be trusted when its trust degree is not lower than 0.5, otherwise it cannot be trusted.

2. Calculate the historical trust value of the smart device based on the historical data. Find the historical data in the server and obtain the communication characteristic value of the current access device in the past k unit time;

Calculate the k nearest-neighbour centroids of the device’s historical communication points according to Eqs. (4)–(6);

Calculate the device historical trust value T2 according to Eqs. (7)–(9).

3. Calculate the smart device trust value and assess the trustworthiness. Calculate the total smart device trust value based on T1, T2 as shown in Eq. (11):

(11) T=T1+T22.

The range of the total device trust value is [0,1], when the trust value is lower than the threshold value of 0.5, the system considers this device untrustworthy and does not allow it to continue communication.

The continuous trust evaluation platform constructs a dynamic outlier detection model based on the trust degree calculation method of centre point drift, continuously evaluates the trustworthiness of the access device, and transmits it to the security protection system. The security protection system detects the trustworthiness value in real time, and if the trustworthiness is lower than the security threshold, it immediately disconnects the communication and controls the abnormal equipment in order to improve the sensitivity of abnormality detection and the level of communication security.

Especially, to balance trust assessment accuracy with system performance, we propose a mechanism for dynamically adjusting the granularity of trust assessments. we classify devices into different categories based on their importance and risk levels, and implement tailored trust assessment schemes accordingly. For high-importance devices, we employ a fine-grained trust assessment approach, which continuously monitors their trustworthiness and detects any potential abnormal behaviors in real time. In contrast, for devices of lower importance, we adopt a coarse-grained assessment strategy, incorporating an incremental evaluation mechanism as well as a trust level caching mechanism. Initially, a trustworthiness value is assigned to each device upon its first access, and this value is cached for subsequent use. The reassessment process is only triggered when there is a change in the device’s behavior. This approach effectively reduces resource consumption while maintaining an optimal balance between assessment accuracy and performance.

Access control engine

Access control mechanism is an important measure to maintain data security and order. Currently, most of the access control mechanisms in substations are static, generating device access policies based on the access requests and security verification results of smart devices and restricting them from accessing the IoT management platform according to fixed permissions. However, due to the dynamic nature of contextual characteristics of power IoT, dynamic access policies are needed to meet its dynamic access characteristics. Static access control mechanisms cannot meet this need.

To achieve dynamic access control, the proposed method builds an access control engine. The engine generates real-time access policies and dynamically enforces them based on the results of device authentication and continuous trust evaluation to achieve dynamic access control. The access control engine consists of a policy administrator (PA), a policy engine (PE) and a policy enforcement point (PEP). Among them, PA is mainly re-sponsible for real-time monitoring and control of communications by issuing commands to relevant PEPs, PE is mainly responsible for generating real-time access policies, and PEP is mainly responsible for enabling, monitoring, or terminating the communication connection between the subject and the object.

The working mechanism of the access control engine is shown in Fig. 5, and the specific steps are as follows:

Figure 5 Working mechanism of access control engine.

1. PA establishes communication with PE to generate access policy. First, after PA obtains the access request forwarded by the local device agent in the data plane, it establishes communication with the policy engine PE, forwards the device access re-quest, and waits for the policy generation. Then, the PE establishes communication with the unified identity management platform to obtain the smart device identity authentication results, i.e., whether the current device access data is complete and whether the device identity is legitimate. At the same time, the PE establishes communication with the continuous trust assessment platform to obtain the real-time trust value of the smart device. Finally, the PE generates a real-time access policy based on the authentication and trust assessment results: when the device identity is authenticated and the trust level is higher than the threshold, the communication is allowed, otherwise the communication is denied or interrupted and transmitted to the PA;

2. PA establishes communication with PEP and dynamically enforces the access policy. Firstly, PA sends corresponding commands to PEP according to the real-time access policy. Then, PEP executes various kinds of instructions issued by PA, including establishing subject-object communication sessions, rejecting subject-object session re-quests, blocking established sessions, etc. After PEP completes the instruction execution, it feeds back instruction execution information to PA;

3. PA continuously communicates with PEP and PE to perform continuity trustworthiness assessment of the session. PEP continuously monitors the session, obtains real-time communication data of the accessing subject, and continuously feeds back communication data and communication parameters of the accessing device to PA. Then, PA, PE, and PEP continuously generate real-time access policies and execute them according to the working mechanism of process (1) and (2) to achieve dynamic access control.

Dynamic access control mechanism monitors communications in real time, generates real-time access policies and enforces them according to the results of identity authentication and real-time trust assessment, and carries out access control in a dynamic way, which can meet the dynamic access characteristics of the Internet of Things in electric power.

Application programme

The application of the proposed method to power systems is confronted with several key challenges, which are outlined as follows: (1) the potential incompatibility of legacy devices with the proposed framework; (2) the high costs associated with continuous trust evaluation for each device; (3) the limitations in scalability imposed by the utilization of the SM9 state secret algorithm; and (4) the inherent complexity of implementing a zero-trust model in the context of real-time operations of power systems. Consequently, this section provides a detailed discussion of these issues.

Device compatibility program

The following solution is proposed to address the first challenge. We distinguish between two types of devices within the grid: modern smart devices and legacy devices. Modern smart devices are capable of supporting the deployment of agent detectors and acquiring the corresponding data, whereas legacy devices lack such support. These legacy devices are not employed as the primary end devices within our framework. Instead, agents are deployed at the upstream edge IoT nodes, which are responsible for collecting the relevant data. Consequently, legacy devices are assigned a relatively lower importance level in the system, given that the accuracy of the data obtained indirectly through the edge IoT agents may not be as high. This differentiated approach facilitates the compatible implementation of the zero-trust model while accommodating both new and legacy devices.

Dynamic mechanism for trust

To address the second challenge, we propose a dynamic mechanism for adjusting the granularity of trust assessment in order to effectively reconcile the trade-off between the accuracy of trust evaluations and overall system performance. Specifically, we introduce a multi-tiered categorization framework based on the relative importance and potential risk of devices within the system. For devices classified as critical or high-risk, we employ a fine-grained trust assessment approach. These devices typically perform essential functions within the system, thereby necessitating a more detailed and comprehensive evaluation of their trustworthiness. A fine-grained assessment involves continuous monitoring of the devices’ reliability, as well as real-time detection and analysis of any anomalous behaviors that may arise. For instance, in the case of high-priority devices, we conduct ongoing trust assessments based on behavioral analysis, which allows for the timely identification of deviations from normal operational patterns and the dynamic adjustment of their trustworthiness. In this process, in addition to leveraging traditional assessment metrics, we also integrate contextual, environment-aware data to enhance the accuracy of the evaluations.

In contrast, for devices of lower criticality, trust assessment can be approached with a coarse-grained strategy that incorporates both an incremental evaluation mechanism and a trust-level caching technique. These devices typically perform non-core functions, and their impact on the overall security and stability of the system is comparatively minimal, thus obviating the need for frequent and exhaustive assessments. Upon their initial interaction with the system, each device is assigned an initial trustworthiness value, which is subsequently cached for future reference. Reassessment is only triggered when a substantial change in the device’s behavioral pattern is detected. Furthermore, the trust-level caching mechanism allows for the storage of assessment results, facilitating faster trust evaluations during subsequent interactions and thereby reducing the consumption of system resources. By adopting this flexible evaluation strategy, we not only optimize system efficiency but also achieve a more favorable balance between the accuracy of trust assessments and the resource overhead required for such evaluations.

Embedded authentication module

To address the third challenge, we propose the design of an embedded authentication module that allows for flexible replacement and customization of authentication mechanisms, adapting to diverse security requirements and application contexts. The module utilizes the SM9 cryptosystem, which employs a computationally complex asymmetric encryption algorithm based on binary linear pairings. This not only enhances identity protection and provides strong defense against attacks but also eliminates the need for traditional digital certificate exchanges. The SM9 design simplifies key management by generating user key pairs based on basic parameters, reducing both the complexity and computational overhead of the authentication process. This results in improved security and efficiency, particularly in large-scale systems.

Furthermore, the modular design of the embedded authentication system allows for greater flexibility and customization in the selection of the identity authentication scheme. In real-world applications, users can tailor the authentication mechanism to specific requirements and scenarios. For instance, encryption algorithms such as RSA (Burkhardt et al., 2023) and AES (Aharoni et al., 2023), which are widely utilized in practice, can be chosen based on the system’s security needs and computational capabilities. This flexibility ensures the system can quickly adapt to different encryption standards and authentication protocols, thereby enhancing its compatibility and scalability. In performance-critical environments, the RSA algorithm, known for its computational efficiency, may be preferred, whereas in scenarios demanding superior data security, the AES encryption algorithm offers enhanced protection. Thus, the embedded authentication module not only optimizes the security and efficiency of the authentication process but also enables dynamic configuration and optimization tailored to diverse operational conditions and user requirements. This approach effectively addresses the trade-off between security and performance across a broad range of application scenarios.

The complexity of implementing the zero-trust

In response to the fourth challenge, we describe below the complexity of implementing the zero-trust solution, focusing our analysis on both technical implementation and deployment, and propose a step-by-step implementation strategy.

In the technical implementation of zero-trust architectures, the integration of key components such as authentication, continuous trust assessment, and dynamic access control policies presents considerable challenges in terms of technical complexity, as well as time and resource consumption. Techniques such as multi-factor authentication enhance security; however, they also introduce significant trade-offs, including increased authentication latency and higher computational resource demands. The process of continuous trust assessment entails real-time analysis of multidimensional data, which further exacerbates the consumption of system resources and increases response latency. Conversely, dynamic access control policies necessitate the continuous adjustment of access permissions based on evolving real-time risk profiles. This dynamic adjustment imposes additional computational and bandwidth burdens, resulting in performance overheads, particularly in complex or highly concurrent environments, and can lead to access delays. To mitigate these challenges and enhance system efficiency, we propose the adoption of trust caching mechanisms and the dynamic adjustment of trust assessment granularity, among other strategies, to optimize resource utilization and reduce latency. To assess the effectiveness of the proposed method, we present a series of experiments designed to evaluate its latency in the subsequent experimental section. Regarding the deployment of the zero-trust framework, we propose a deployment strategy that ensures compatibility with both newer and legacy devices. This approach eliminates the need for large-scale modifications or updates to existing devices, thereby minimizing implementation complexity. Furthermore, we introduce a phased implementation strategy. This strategy advocates for the gradual adoption of zero-trust principles, beginning with selected subsystems and progressively expanding to encompass the entire network. By adopting this incremental approach, we aim to reduce the complexity associated with the initial deployment and facilitate the resolution of potential issues in a stepwise manner.

Experiments and analyses

In order to verify the practical application effect of the proposed designed illegal device detection scheme based on zero-trust architecture, the usability of the system needs to be tested. The access scenario is shown in Fig. 6: on the terminal side, multiple collection terminals access the edge IoT agent system, and then the edge IoT agent forwards the device communication request to the security protection system through the device agent module. After the illegal device obtains the request by the detection system, it carries out identity authentication and continuous trust assessment of the device, and dynamically controls the secure communication between the IOT agent and the scheduling D5000 system according to the access policy. In Fig. 6, APN is the private network and MCDN is the Mobile Content Delivery Network.

Figure 6 Experimental environment.

In particular, the deployment of the proposed system is specified as follows. A gateway server is developed based on the proposed system, and this server is deployed on the front side of the property management platform. The efficacy of the continuous trust assessment module and the dynamic access control module is executed when business communication is performed within the power system. The performance of the gateway server is better to support continuous trust assessment and detection and perform fine-grained dynamic access control. For normal terminals, business execution is allowed. And for abnormal terminals, they are not trusted and access policies such as blocking communication are executed.

The experimental setup is shown in Table 2. In the experiment, 10 acquisition devices are set, and each device continuously up-loads data in 10 unit times. A unit time is set to be 5 min, and the time interval of collecting data in each unit time is 30 s. The raw data of the edge IoT agent acquiring devices contains device identification and communication characteristic value.

Table 2 Experimental setup.

Norms	Values	
Acquisition device	10	
Continuously upload data	10 unit times	
One unit time	5 min	
Time interval of collecting data in each unit	30 s	
Collect data	Device identification and communication characteristic	

Security detection effectiveness experiment

The experimental test index is the security verification results, and the test results are shown in Table 3. A total of 100 groups of samples are set for the experiment, including 26 groups of abnormal data. According to the experimental results, the empirical parameters are constantly revised. When the empirical parameter takes the value of 0.22, the anomaly detection rate is 96.15%, and the security verification accuracy rate is 98.6%. The experimental results show that by selecting the appropriate empirical parameter, the security detection of the illegal device detection system for access devices is more accurate, and the detection model is less affected by the abnormal values.

Table 3 Results of security detection.

Sample size	Anomaly sample	Empirical parameter	Successful detection	
100	26	0.1	10	
100	26	0.2	22	
100	26	0.3	18	
100	26	0.25	21	
100	26	0.22	25	

Time delay experiment

The real-time generation of dynamic access policies can introduce delays, particularly in large Internet of Things (IoT) networks, where real-time responsiveness is of paramount importance. In critical infrastructures such as power grids, even minimal delays can precipitate significant operational issues. To assess whether the zero-trust framework contributes to such delays, we compare our proposed approach with the traditional Attribute-Based Access Control (ABAC) model. Specifically, we evaluate the average decision-making time of both models, as illustrated in Fig. 7, and the accuracy of their interactions, as shown in Fig. 8. The experimental results indicate that our approach achieves a notably higher interaction correctness rate compared to the traditional ABAC model. While trust evaluation plays a central role in our model, directly influencing its overall performance, the decision-making time in our model is slightly longer than that of the ABAC model. However, as the volume of data increases, the time delay tends to stabilize, and this delay remains within acceptable limits.

Figure 7 The comparison of average decision time.

Figure 8 The comparison of average correct interaction rate.

Conclusion

We analyse the security status of the new electric power system and the research status of the application of zero-trust architecture in the power field. Aiming at the security risks faced by the “once-authentication, always-trust” mechanism, a scheme for detecting illegal terminals in IoT based on zero-trust architecture is proposed. The proposed system follows the zero-trust concept of “never trust, always authenticate”. Firstly, the identity of the smart device is verified for legitimacy. Then, continuous trust assessment is performed for legitimate devices to generate real-time assessment values. Finally, real-time access policies are generated and dynamically controlled to ensure communication security. The results of security detection experiments show that the detection effect of the system is more dependent on the selection of empirical parameters, and when the empirical parameters are appropriate, the security detection results of the system on the access device are more accurate and less affected by the outliers. Therefore, the next work direction is how to effectively correct the empirical parameters to quickly obtain stable system detection results.

Additional Information and Declarations

Competing Interests

Hongyu Zhu, Jianwei Tian & Zheng Tian are employed by State Grid Hunan Electric Power Company Limited Information and Communication Company and Hunan Key Laboratory for Internet of Things in Electricity. Qian Chen & Weiqiang Luo are employed by State Grid Hunan Electric Power Co., Ltd. Mingguang Li is a student at Hunan University.

Author Contributions

Hongyu Zhu conceived and designed the experiments, performed the computation work, prepared figures and/or tables, authored or reviewed drafts of the article, and approved the final draft.

Jianwei Tian conceived and designed the experiments, prepared figures and/or tables, and approved the final draft.

Qian Chen conceived and designed the experiments, analyzed the data, prepared figures and/or tables, and approved the final draft.

Zheng Tian performed the experiments, analyzed the data, authored or reviewed drafts of the article, and approved the final draft.

Weiqiang Luo performed the experiments, analyzed the data, performed the computation work, authored or reviewed drafts of the article, and approved the final draft.

Mingguang Li conceived and designed the experiments, performed the experiments, performed the computation work, authored or reviewed drafts of the article, and approved the final draft.

Data Availability

The following information was supplied regarding data availability:

The data is available at Zenodo: LI, M. (2024). Zero-Trust [Data set]. Zenodo. https://doi.org/10.5281/zenodo.14402857.

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
