# Peer review of "A zero-trust based scheme for detecting illegal terminals in the Internet of Things of smart grid"

_PeerJ Computer Science, doi:10.7717/peerj-cs.2736_

## Round 0.1 · original submission · Major Revisions

Dear authors,

You are advised to critically respond to all comments point by point when preparing an updated version of the manuscript and while preparing for the rebuttal letter. Please address all comments/suggestions provided by reviewers, considering that these should be added to the new version of the manuscript.

Kind regards,
PCoelho

Reviewer 1 ·

Basic reporting

Generally, the proposed solution is described in a clear and unambiguous English language. However, improvements on the academic English writing needs to be done. Please consider the following points on few suggestions:

- The academic writing needs to be improved, as in many places the text is written in poor unacademic English language. For example, in line 55, rephrase or use "However" instead of "but" ".

- Avoid writing very long paragraphs. Many paragraphs in the text need to be shortened, partitioned or rephrased, please check the uploaded annotated pdf document for places on where you need to do the change.

- In many places I read the phrase "Illegal device detection scheme". Although I understand the context, but the way it reads, doesn't sound right. Sometimes it read as if the detection scheme itself is illegal. Again, I understand the context, but I would prefer to rephrase, for e.g., by writing " a scheme for detecting illegal devices etc.."

In the following I provide just a suggestion on how to tackle the above points, to improve the paragraph from line 54-59:

"The above research on detecting illegal devices in power IoT ensures the legitimacy of the access
device. However, such solutions impose serious security risks as the nature of their authentication mechanism is "once authenticated, always trusted ". In this paper, we propose a scheme for illegal device detection based on zero-trust architecture. The proposed scheme, aims to authenticate all the accessed smart devices and conduct continuous trust assessment to achieve continuous supervision and dynamic access control of the session."

- When referring to a literature work, the most common way is to write: the work proposed in [] or authors et al [] did this, but not literature []. I suggest you re-write how you refer to a work proposed in literature.

-In many places in the manuscript I found repeated paragraphs. I annotated some on the uploaded PDF. Please rewove the repetition and revise elsewhere.

- The introduction section can be improved by providing more background on what the power IoT is, and provide an example about its structure, and a realistic figure about the system, so the reader can get a better idea of the risk and the impact of cyber-attacks on such systems. Moreover, elaborate more on the concept of the Zero Trust Architecture. Additionally better to elaborate more on the problem, and the suggested solution in this section.

- I am not sure if the authors are following specific a PeerJ template structure, but as a suggestion, I find it better to have a specific section for the related work. I mean moving the text from line 41-53 from the Introduction and merge it with the literature work presented in Section "Current Research Status of Electric Power Network Security Solutions Based on Zero Trust Architecture " and put them under one section called related work or literature work.

- The text in the figures is read clearly. However, the way the blocks are done and the notations provided are confusing, especially in Figs 1-3. For example, in Figure 1, its not clear whether the proposed solution of the data and control plane is implemented inside the terminal? or in which part of power IoT system, the solution is be implemented? Also, the blocks in Fig 2 and Fig 3 and the notations provide in Fig 2 are very confusing. It is better if you present the proposed scheme in a clearer way and make the notations clearer by e.g. using a flow chart.

-Moreover, few abbreviations are provided in the figures, such as in Fig 4, it’s better to mention what is MCDN and APN refer to, so more audience can understand what they refer to.

- Please refer to the uploaded annotated PDF for more comments on the text to consider.

Overall, It could be useful if the authors ask a colleague who is proficient in English academic writing and familiar with the subject matter to review the manuscript.

Experimental design

The knowledge gap being investigated is clearly identified and the proposed solution contributes to filing the gap. However, there are unclear /confusing points that needs to be clarified.

- For example, in which part of the power IoT system the proposed scheme will be implemented? It is important to answer this question in order to evaluate the performance level of the proposed scheme, given that the terminals usually are resource constrained devices, and running the scheme seems to require rich resources.

- More details need to be provided about the conducted experiments, e.g., devices used as terminals, and as an edge, details about the environment such as the communication medium used, etc. because those details would affect the trust assessment results in terms of communication delay and packets ratio etc. It is good to know about such details used for the experiments, especially that Fig 4 doesn't explain well the experimental setup.

- It’s better to describe the experimental/simulation setup in a table form.

- More use case scenarios need to be conducted to verify the accuracy of the proposed solution. For example, the results show cases that indicate false negative performance of the solution depending on the empirical parameter value, besides that, it is good to see whether there are cases for false positive, to determine the level of the accuracy of the scheme.

-It would better to compare the performance of the proposed technique with the performance of other related techniques proposed in literature, to understand better the usefulness of the technique over other techniques.

Validity of the findings

- The performance of the proposed solution, along with its correctness and accuracy are not validated. It will be good if the performance of the proposed technique is evaluated by comparing it with other techniques from literature, under the same environmental and simulation setup, so to know why one can adopt the proposed solution for detecting the illegal devices and not other solutions.

-The proposed solution is not validated under different conditions that can affect the trust assessment, such as the communication medium, technologies and devices used etc.

- The proposed solution does not consider the performance and efficiency of the algorithm especially running on resource constrained terminals.

- A very important point to consider, how robust is the proposed scheme itself against cyber-attacks. Moreover, how the proposed solution can tolerate the cyber threats adaptively? or whether there are backup plans considered? good to consider this as a future work if it is difficult for this work.

- As it is explained above, the experiments do not seem to have considered cases for false positive outcome, which is important for indicating the validity and the accuracy of the proposed solution.

Additional comments

Overall, the proposed solution is good but needs to consider the comments above, or at least those suggested in the basic reporting section, besides considering the comments provided in the uploaded annotated PDF document.

Annotated reviews are not available for download in order to protect the identity of reviewers who chose to remain anonymous.

Reviewer 2 ·

Basic reporting

The research proposes a zero-trust based illegal terminal detection scheme for the Internet of Things (IoT) in electricity networks, focusing on protecting against compromised terminals. While the system's design, based on continuous trust evaluation and dynamic access control, shows promise, there are some challenges and areas for improvement. Below are potential problems and suggestions:

1. The Complexity of Zero-Trust Implementation is not detailed in the manuscript. The authors should talk about the complexity of the zero-trust implementation.

2. IoT networks, especially in electricity grids, involve a massive number of devices. The continuous trust evaluation for each device could cause scalability problems, leading to bottlenecks in the detection process or delays in access decisions. How do the authors address this?

3. Many electricity grids still rely on older legacy systems and devices that might not be compatible with advanced zero-trust-based schemes. What are the costs and challenges of retrofitting these systems with new security mechanisms?

4. Using the state secret SM9 system for authentication is promising but specific to some cryptography standards. This could limit its global applicability, as not all regions may adopt or trust the SM9 cryptographic system. Is there any validity for this?

5. Most of the in-text figures and tables are missing. The authors need to double-check the paper.

6. The references are not adequate. The authors should include more relevant and updated references in the revised version. It is also suggested that they draw a table to summarize the comparison with related work.

7. The simulation results are not sufficient for the current version. More results are needed to strengthen the use of the proposed methods. Specifically, the real-time generation of dynamic access policies may introduce latency, especially in larger IoT networks where real-time responses are critical. In electricity grids, even minor delays can cause significant operational problems.

8. To increase the SM9 system's global applicability, consider supporting multiple cryptographic standards in addition to the SM9 system. Supporting common standards like RSA, AES, or ECC could make the system more versatile across different regions and industries.

Experimental design

Please see the above comments.

Validity of the findings

Please see the above comments.

Additional comments

Please see the above comments.

---

## Round 0.2 · Major Revisions

Dear authors,

You are advised to critically respond to all comments point by point when preparing an updated version of the manuscript and while preparing for the rebuttal letter. Please address all comments/suggestions provided by reviewer #2 in the previous round (check the annotated manuscript), considering that these should be added to the new version of the manuscript.

Kind regards,
PCoelho

Reviewer 1 ·

Basic reporting

All is good. The authors considered all my comments.

Experimental design

All is good. The authors considered all my comments.

Validity of the findings

All is good. The authors considered all my comments.

Reviewer 2 ·

Basic reporting

I checked the response letter and revised the manuscript thoroughly. The authors did not incorporate any of my comments. Therefore, I must reject this paper and suggest the authors respect the reviewers' time. If the authors appeal to this decision, I must say to highlight all the changes according to my comments in RED colour.

Experimental design

See the comments in the basic reporting.

Validity of the findings

See the comments in the basic reporting.

Additional comments

See the comments in the basic reporting.

---

## Round 0.3 · accepted · Accept

Dear authors, after consideration, we believe that you have adequately addressed the feedback.

Thank you for considering PeerJ Computer Science and submitting your work.

Kind regards
PCoelho

Reviewer 2 ·

Basic reporting

I have reviewed this paper several times but am not satisfied with the work and revisions. There are still many problems in the paper, and it does not follow the standard format. Also, according to the previous comments, the authors did not revise the paper very well. Therefore, I must reject the paper.

Experimental design

See the comments above.

Validity of the findings

See the comments above.

Additional comments

See the comments above.